# Active boundary layers in confined active nematics

Jerôme Hardoüin[1,2], Claire Doré ®[3], Justine Laurent ®[3], Teresa Lopez-Leon ®[3], Jordi Ignés-Mullol ®[1,2] ✉ & Francesc Sagués[1,2]

The role of boundary layers in conventional liquid crystals is commonly related to the mesogen anchoring on confining walls. In the classical view, anchoring enslaves the orientational field of the passive material under equilibrium conditions. In this work, we show that an active nematic can develop active boundary layers that topologically polarize the confining walls. We find that negatively-charged defects accumulate in the boundary layer, regardless of the wall curvature, and they influence the overall dynamics of the system to the point of fully controlling the behavior of the active nematic in situations of strong confinement. Further, we show that wall defects exhibit behaviors that are essentially different from those of their bulk counterparts, such as high motility or the ability to recombine with another defect of like-sign topological charge. These exotic behaviors result from a change of symmetry induced by the wall in the director field around the defect. Finally, we suggest that the collective dynamics of wall defects might be described in terms of a model equation for one-dimensional spatio-temporal chaos.

In fluid dynamics and soft materials science, boundary layers refer to the part of the flow that is close to a boundary, featuring built-up slip velocities associated to localized profiles of scalar fields, like pressure, temperature, solute concentration or ionic charge density[1,2]. Although the length scale spanned by such boundary layers is much smaller than any macroscopic length, the details of the interfacial transport processes entirely control the distant fluid dynamics. Here, we reveal the existence of a new type of boundary layer in a confined Active Nematic (AN)[3], which we call active boundary layer (ABL).

Defects, i. e. regions of undefined local order, is similarly a transversal concept in condensed, both hard and soft, materials. Although normally transitory for systems near equilibrium, defects play a significant role in many contexts, from phase transitions, mediated by defect proliferation[4,5], to colloid assembly, via defect entanglement in liquid crystal (LC)-based dispersions[6–8].

In this latter context, defects refer to singularities in the orientational field that are endowed with topological indices that bear characteristics similar to electric charges. Stressing this resemblance, and expanding the conventional notion of electrified interfaces, we demonstrate that active LCs[3,9,10] engender ABLs that become polarized by accumulating equal-sign topological defects.

We experimentally realize two-dimensional ANs by the self-assembly of filamentous bundles of microtubules and kinesin molecular motors at the water/oil interface[11–13], which form an active ordered film populated by motile topological defects characterized by semi-integer topological indices, or charges[3]. Trefoil-like -1/2 defects are passively advected by the flow while comet-like +1/2 defects actively drive the system. Experiments[11,12,14–17] and theoretical models[18–22] show that ANs feature a balanced population of positive and negative semi-integer defects that continuously unbind and recombine, resulting in a turbulent mixing regime[16,23–25]. Different from this traditional view, we show here that, under lateral confinement, ANs can feature ABLs that are exclusively populated by negative defects. When analyzed individually, these boundary defects display differences with respect to their bulk counterparts, both in terms of their motility, symmetries, and built-up stresses. Remarkably, we identify recombination events between equal-sign defects residing at the boundary.

[1]Departament de Química Física, Universitat de Barcelona, 08028 Barcelona, Spain. [2]Institute of Nanoscience and Nanotechnology, Universitat de Barcelona, 08028 Barcelona, Spain. [3]Laboratoire Gulliver, UMR CNRS 7636, ESPCI Paris, PSL Research University, 75005 Paris, France. ✉e-mail: jignes@ub.edu

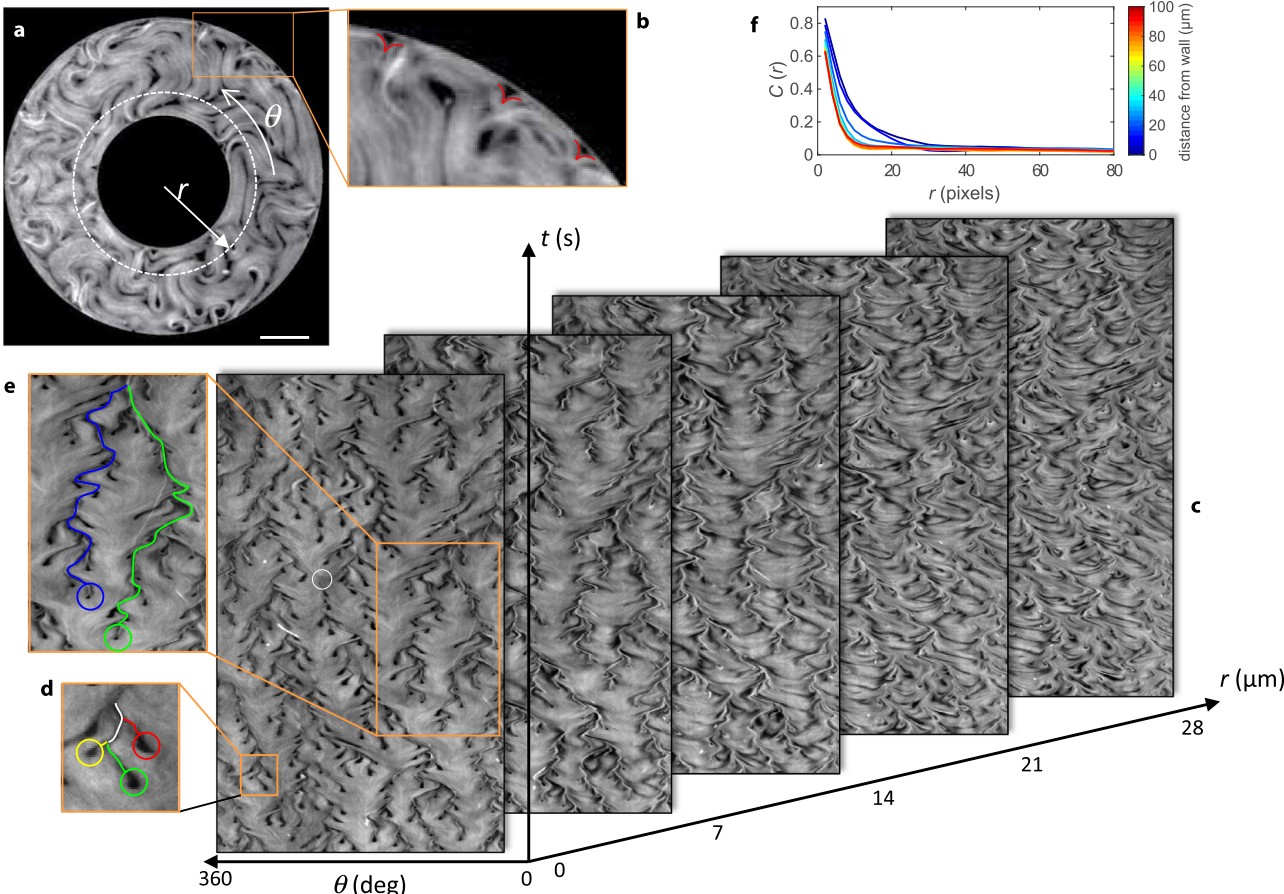

**Fig. 1 | Active nematic dynamics close to a wall. a** Fluorescence micrograph of the active nematic flowing inside an annular channel (see Supplementary Movie 1). Scale bar: 100$\mu$m. **b** Magnified image showing three of the −1/2 defects that form the outer ABL. **c** Kymographs corresponding to the fluorescence intensity profile along circumferences ($\theta$ spanning from 0 to 360 degrees) at increasing distances from the inner channel wall, $r$, as indicated in **a**. We consider the inner wall to be at $r = 0$. For each plot, circular profiles at different times are stacked vertically, for a total duration of 300 s. The white circumference in the $t = 0$ kymograph highlights a rare event of a boundary defect annihilating with a bulk defect, as explained in the text. **d** Region of the inner-most kymograph showing three blossoms that merge into a single branch. **e** Region of the inner-most kymograph showing the long-range attraction between branches. **f** Radial autocorrelation of the kymograph images, $C(r)$, for kymographs obtained at different distances from a flat wall, $r$.

Under strong confinement, the structure and dynamics of the ANs within the boundary is preserved and it even enslaves the bulk organization. For instance, in small disks, a single wall defect is responsible for ejecting streaming jets that determine the system-wide flow. Finally, by going from the individual to the collective dynamics of boundary defects, we show that their concerted effect echoes a purely one-dimensional Kuramoto-Sivashinsky (KS)-like description of spatiotemporal chaos[26].

## Results

### Self-assembly of active boundary layers

To characterize the ABLs, we use annular shaped channels, a geometry that allows us to avoid end effects. The channels are sufficiently large to ensure that the dynamics of the ABLs at either wall are effectively decoupled. A few microliters of the kinesin/tubulin active mixture is placed in a custom-made open pool 5 mm in diameter, and subsequently covered with 100 cSt silicon oil (see Methods)[12,13]. Confining micro-platforms are set in contact with the oil/water interface once the AN is formed. They are flat, rigid polymeric slabs with pierced micro-channels manufactured with three-dimensional micro-printing techniques (see Methods).

The used micro-channels induce lateral confinement of the quasi-two-dimensional AN, and impose planar anchoring conditions on the active filaments and favorable slip velocity conditions along the channel walls[13].

In Fig. 1, we show data for an annulus of 200 μm width. In the absence of confinement, the used material parameters for the AN lead to an average defect spacing of 100 μm, but defect density is increased by the lateral confinement inside the channel[13]. Fluorescence micrographs display the typical active turbulence regime away from the walls (Fig. 1a). The latter become preferred sites for defect nucleation[13,27–29] since aligned extensile ANs are prone to bend-like instabilities[25,30,31] that lead to the unbinding of defect pairs[11,20]. In the process, +1/2 defects are ejected from the walls into the bulk, while negative counterparts remain at the boundary (Fig. 1b). As a result, a new structure emerges near the boundary, whose dynamics is controlled by exotic topological defects that nucleate, move and annihilate at the wall following unusual dynamics.

To better analyze the process, we have built kymographs by measuring the fluorescence intensity along circumferences parallel to the boundary and we have stacked vertically the resulting pixel lines obtained at increasing times (Fig. 1c).

Close to the inner wall ($r \simeq 0$), kymographs appear anisotropically patterned, with tree-like lines, separated by smoother regions of relatively uniform intensity. Branches originate from dark blossoms that pop up from uniform regions, and correspond to spontaneous events of boundary defect nucleation, while branching points correspond to defect merging events (Fig. 1d). Branches are often short, corresponding to ephemeral defects, although the existence of long-lived branches are the signature of resident wall defects that

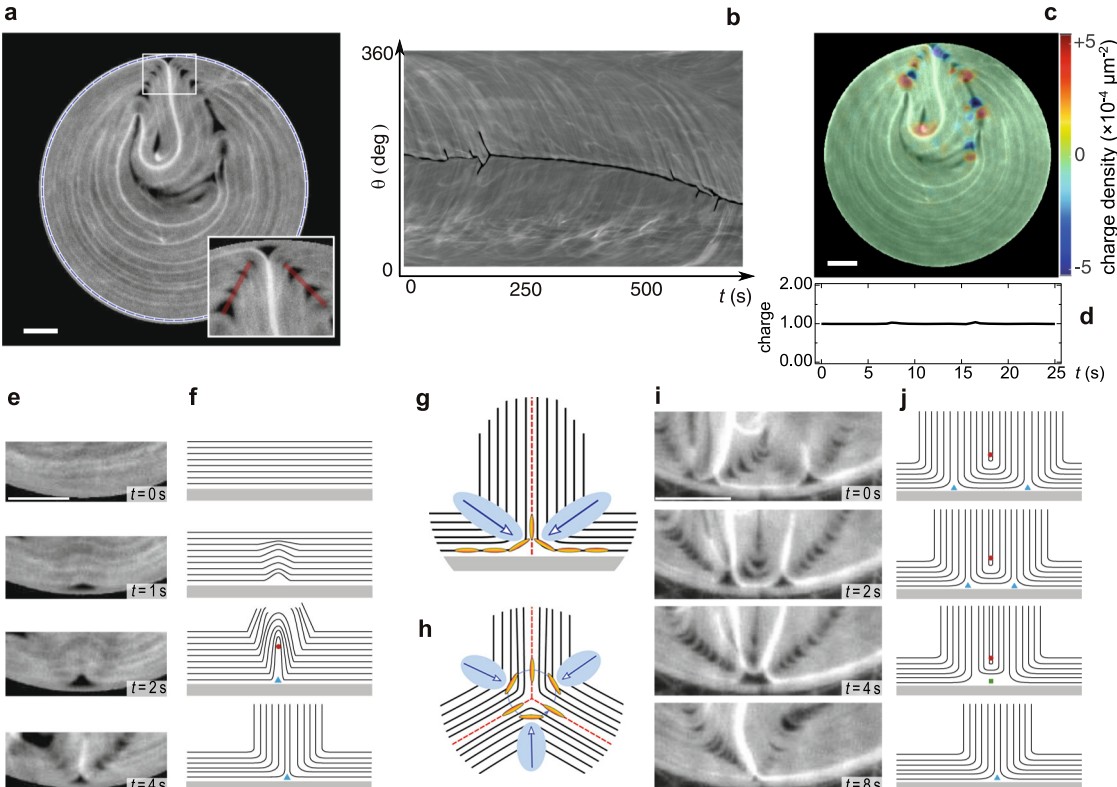

**Fig. 2 | Birth and death of wall defects. a** Fluorescence micrograph of the active nematic evolving inside a circular pool (see Supplementary Movie 2). In the inset, a magnified view of the region around the single wall defect showing the bright plume and highlighting in red the lateral crimps (see text). **b** Kymograph built with the grayscale intensity along the blue dashed circumference in **a**, as described in Fig. 1. **c** Local charge density of the director field for the image in **a**. Charge density is computed as detailed in the Methods, and it is overlaid as a color map on the fluorescence image (see Supplementary Movie 3). **d** Topological charge integrated over the full disk-shaped pool. **e** Experimental time-lapse of a wall-defect nucleation in the disk. **f** Corresponding sketch of the dynamics. Structure and active forces of a wall (**g**) and a bulk (**h**) −1/2 defect. The determination of the topological charge is shown for the bulk defect. Red dotted lines indicate the symmetry axes and blue areas are the regions with higher active stress. Blue arrows mark the direction of the resulting active force. **i** Experimental time-lapse of the merging between two wall defects in the same system. (j) Corresponding sketch of the dynamics. In **e-j**, the confining wall is located at the bottom. Scale bars, 100 μm. In the sketches, red circles denote the core of a +1/2 defect, blue triangles that of a −1/2 defect and the green square, that of a −1 defect.

experience long-distance attraction (Fig. 1e). This hierarchical dynamics gradually vanishes away from the wall. Indeed, we find that kymograph patterns are highly correlated close to the wall but they become structureless around 25 μm away from the wall (Fig. 1f). This penetration length for the ABL is likely to depend on the active length scale, $\ell_\alpha$, which is determined by the composition of the AN, and also on the channel width, which competes with $\ell_\alpha$ to determine the geometry of AN under confinement[13]. A qualitatively similar dependence on activity of the anchoring penetration depth has recently been reported in simulations of confined active fluids under external forcing[32]. The situation is similar at the inner and at the outer annulus walls, and also near flat walls in rectangular channels (Supplementary Fig. 2), indicating that ABLs form regardless of the channel geometry.

## Structure and dynamics of wall defects

Isolated wall defects are best tracked in disk-shaped pools, where the AN displays rather uniform textures and system-wide quasi-laminar flows (Fig. 2a). This is an indication of the presence of very few wall defects, often a single one for most of the observation time, with the occasional emergence and quick annihilation of additional wall defects. The corresponding kymograph of the boundary texture displays scarce blooms and branches (Fig. 2b). As observed in the kymograph, the single wall defect can either fluctuate around a random fixed position for an undetermined period of time (first 250s in Fig. 2b and Supplementary Fig. 3a) or drift autonomously along the circular boundary, clockwise or counter-clockwise handedness having

similar likelihood (from 250s onwards in Fig. 2b and Supplementary Fig. 3b). Occasionally, this dynamics is interrupted by brief episodes of wall defect nucleation and annihilation, (Supplementary Fig. 3c), in a process that we describe in detail below.

As a result of the planar boundary conditions on the circular boundary, the total topological charge of the confined AN is expected to be +1[33,34], regardless of the complex defect dynamics described below. To validate this expectation, we have first computed the local charge density as proposed in ref. 35 (see "Methods"). The charge density is shown in Fig. 2c for the same fluorescence micrograph in Fig. 2a (see also Supplementary Movie 3). By integrating the charge density over the entire surface, we obtain a constant value +1, as expected (Fig. 2d).

The birth of a wall defect is shown as a time-lapse image sequence in Fig. 2e, while the structural changes associated to the process are sketched in Fig. 2f. The extensile filaments aligned with the boundary become unstable with respect to bend instabilities. This leads to the unbinding of semi-integer defects (see third panel in Fig. 2e). The cores of the +1/2 and −1/2 defects are marked with a red circle and a blue triangle, respectively, in Fig. 2f. Notice that this defect-creation event incurs in no net topological charge change in the system. While the positive defect is ejected into the bulk, the core of the −1/2 defect remains at the wall. Confocal microscopy images (Supplementary Fig. 4) reveal that the core of the defect, devoid of active filaments, sits on the wall. This is consistent with the fact that wall defects are not injected into the bulk, which would occur if active

filaments, prone to bend instabilities, would be present between the core and the wall.

Wall defects feature a two-fold symmetry about an axis perpendicular to the boundary (Fig. 2g), which is different from the three-fold symmetry of bulk −1/2 defects (Fig. 2h). The defect core is prolonged by a plume of high-density fluorescent filaments (Fig. 2a), reminiscent of the beating active filaments reported by Sanchez et al. at the boundary of a bulk active gel[36]. Inspection of the quasi-laminar flow profiles inside the pool (see Supplementary Movie 2) reveals two steady counter-rotating vortices symmetrically organized on either side of the wall defect with velocities parallel to the pool boundary. These flows converge at the tip of the wall defect, effectively concentrating the fluorescent active filaments, which results in a bright plume that is advected by the flows.

The wall defect is surrounded on both sides by active filaments whose bend distortion is higher than in the case of bulk −1/2 defects (Fig. 2g, h), resulting in higher elastic stresses. Because of their rigidity, filaments cannot accommodate the higher curvature, and additional void regions appear as crimps that steadily flank the wall defect (Fig. 2a).

A decrease in the number of defects in the ABL typically proceeds through the merging of equal-sign wall defects, facilitated by the intercalation of a +1/2 defect that appears as the two −1/2 defects get closer (see Fig. 2i, j; see also Supplementary Figure 4). Eventually, the two −1/2 boundary defects merge to form a −1 boundary defect, which quickly recombines with the extruded +1/2 defect as the latter is compressed beyond the bending ability of the active filaments (see Supplementary Movie 4).

On the other hand, direct recombination of a single wall defect with a bulk +1/2 defect, although rare, is also possible. This type of event manifests itself in the kymographs as branches that suddenly end, without connecting to another branch. An example is highlighted with a white circumference in Fig. 1c, and is also visualized in high resolution, near a flat wall, in Supplementary Fig. 4 and in Supplementary Movie 4.

### Pinning of wall defects and entrainment of bulk active flows

Bulk active flows can be controlled by direct intervention in the structure of the ABL. This is achieved after modifying the normally smooth boundaries to introduce indentations much smaller that the typical coherence length of the ANs, in order to anchor wall defects at predefined locations. Here, we demonstrate this with disk-shaped pools using single indentations that are ten times smaller than such typical length scales (around 100 μm[25]). This minimal intervention offers remarkable perspectives towards the large scale control of the ANs, as illustrated in Fig. 3 for different disk sizes.

Under the conditions in that experiment, the bulk dynamics is rather chaotic for the largest disk of 400 μm diameter (Fig. 3b). A considerable number of wall defects nucleate randomly, besides the one emerging from the corrugation, as can be seen in the corresponding kymograph of the fluorescence intensity along the boundary perimeter. The average number of wall defects decreases when decreasing the disk size, while the indentation acts as a defect attractor where many dynamic branches collapse (Fig. 3c, g). As the disk diameter reaches 180 μm, the ABL starts to determine the overall flow. In this case, nucleation of additional wall defects is mostly limited to a single defect across the diameter from the corrugation (Fig. 3d), as can be observed in the corresponding kymograph, where only a few additional, short-lived branches are observed (Fig. 3h). Finally, when the diameter is decreased below 130 μm (Fig. 3e, i) a single pinned wall defect is observed. A fluorescent plume, with high concentration of active filaments, constantly emanates from the corrugation, without additional disturbances in the ABL. This leads to periodic oscillations (Fig. 3j, k) of this plume, similar to the cilia-like beating identified for bundled microtubules by Sanchez et al.[36].

### Force balance within the active boundary layer

Before studying collective effects, we pursue the description above in a more quantitative basis by assessing the active forces acting on single and neighboring pairs of wall defects. We first evaluate the tensor order parameter from the nematic director field extracted from fluorescence micrographs[37], and we further compute its local gradients to map the local active forces. In the Methods section we give details on how this protocol is implemented. Figure 4 illustrates the situation for a single defect (Fig. 4a–c) and for a pair of defects prior to their recombination (Fig. 4d-g). In the first scenario, a nearly symmetric force distribution self-organizes along the crimps that emanate from isolated wall defects, protecting them and keeping them confined within the boundary layer (see Fig. 4c). On the other hand, when two wall defects approach (Fig. 4d), the force distribution is no longer symmetric around each defect, and a net attraction between the defects builds up, as observed in the map of the active force component along the boundary (Fig.4e, f). A sketch illustrating this force imbalance is presented in Fig. 4g.

We have also looked for signatures of a coupling between wall defect nucleation events and the evolution of the stress in the whole system. We have found that wall defects nucleate randomly and at random positions, but only when the average tangential stress in the system (parallel to the circular wall) is either a maximum or a minimum in the course of time (Fig. 4h). Interestingly, a different dynamics is observed in the experiments by Opathalage et al.[28] on an AN inside circular cavities. In their experiments, a pair of central evolving +1/2 defects is periodically disturbed by nucleation of a couple of oppositely charged defects. The new positive defect further unbinds and replaces one of the original central +1/2 defects, which annihilates with the newly created negative defect (Fig. 4i). Minima of the stress occur just before defect nucleation. This periodic behavior is reminiscent of the observations in unconfined or weakly confined ANs[15,16,38], and different from our system whose dynamics is controlled by ABLs. While, in our work, lateral confinement is applied on an already-formed AN, experiments in Ref. 28 are performed by assembling the AN on a patterned, oil-coated substrate. The latter results in the formation of menisci along the pool boundaries, which may lead to interfaces that deform along the normal direction, thus preventing observation of the AN with a good resolution.

### Wall defects: collective behavior

In this section, we present a tentative analysis for the collective dynamics of wall defects in ANs, which is based on an analogy with patterns arising from the Kuramoto Shivashinski equation (KSE). The latter is the simplest model describing the transition to spatiotemporal chaos for spatially extended systems subject to an instability at a well-defined length scale[26], and has been applied in the study of pattern formation in a wide variety of condensed matter systems[26,39–41]. Conceptually, this analogy arises because ANs constitute a good example of emergent spatiotemporal chaos, and they feature intrinsic time and length scales[3]. Moreover, it has recently been shown that the transition to active turbulence in ANs can be characterized with a genuine pattern-selection mechanism[25].

The KSE yields the evolution of a one-dimensional scalar field, $\psi(x;t)$. Interestingly, the spatiotemporal patterns emerging from these simulations bear a striking resemblance with the kymographs from the ABL (see Supplementary Fig. 1a, e), even though branches nucleate in nonsingular regions in KSE simulations, as there are no defects in $\psi(x;t)$. It is also not obvious which physical field in the AN should play the role of $\psi(x;t)$ to carry this analogy further. A useful choice is to consider it related to the orientation of AN filaments at the walls.

The most distinctive feature of KSE chaos is its spectrum of spatial fluctuations, considered as a proxy of the energy spectrum. It is mainly characterized by a well-defined peak at short wavelengths followed by a short-range scaling at intermediate wavelengths. In Supplementary

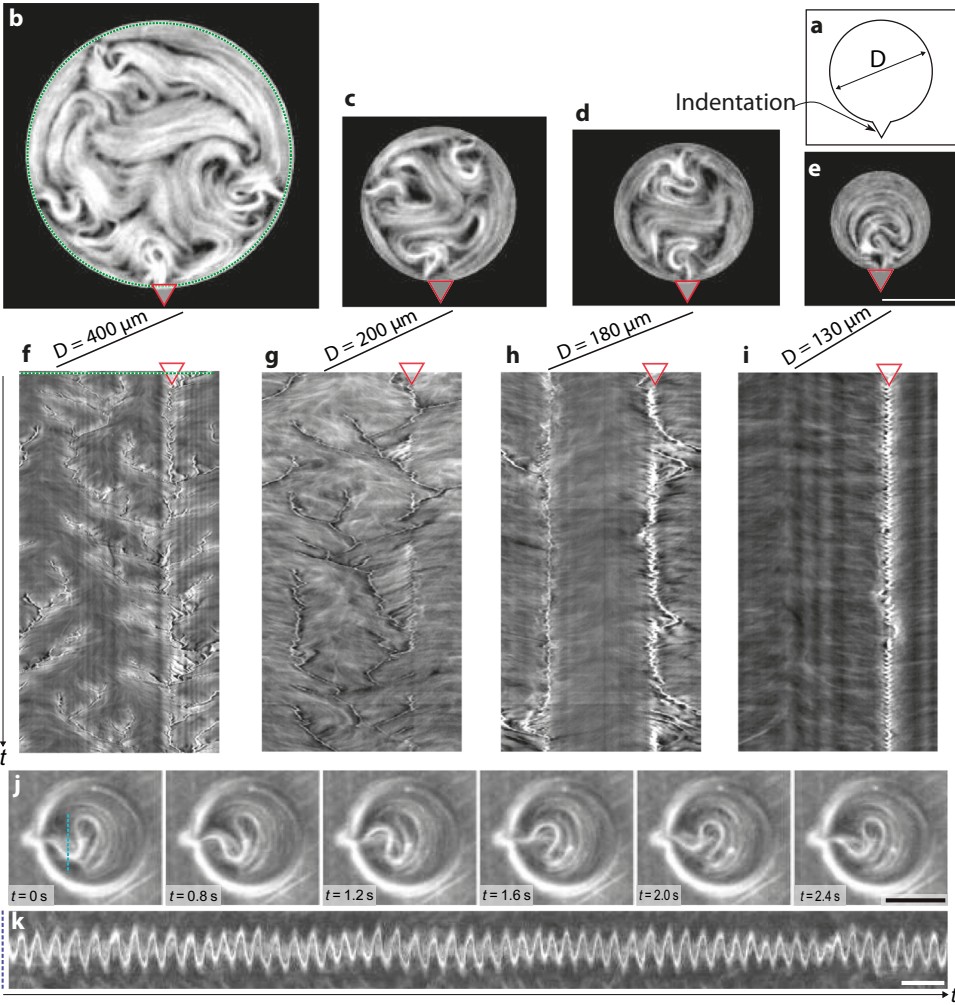

**Fig. 3 | Wall-pinning determines defect dynamics. a** Sketch of the corrugated disk geometry. The indentation is a triangle with a fixed width and height of 10 μm. **b**–**e** Fluorescence micrographs of active nematics confined in corrugated disks with diameters $D$ = 400, 200, 180, 130 μm, respectively (see Supplementary Movie 5). The red triangle locates the indentation. Scale bar: 100 μm. **f**–**i** Space-time plots along the perimeter of each disk. Total elapsed time is 300 s. The red triangles locate the position of the indentation. **j** Time-lapse of the periodic oscillation of active nematics confined in a 130 μm corrugated disk. The indentation is located to the left side of the disk. Scale bar: 100 μm. **k** Space-time plot along the blue dotted line in the first panel in **j**. Scale bar: 10 s.

Notes 1 and 2, we compare the spectra of fluorescence intensity fluctuations along a wall in an annular channel and for the order parameter in one-dimensional KSE simulations. Qualitatively, KSE and experimental AN spectra do show similarities (Supplementary Fig. 1d, h), with the existence of a peak followed by a power-law decay.

Based on the above similarities, we have decided to adapt recent work by Toh[42] to provide a collective description of defects in the ABL. In that work, the framework of the KSE was successfully analyzed in terms of a statistical description of the interactions between short-lived and long-lived soliton-like structures, which here we relate to defects in the ABL (Fig. 5a), and the energy spectrum was calculated as in ref. 42 (Supplementary Fig. 1). In this alternative analysis of kymographs, we begin by discriminating defect trajectories in terms of their persistence, defined as the time between their nucleation and their merging with an older defect; when two defects merge, we consider that the older one is preserved while the younger one is annihilated (Supplementary Fig. 5, Fig. 5b, c). By performing a statistical analysis of these lifetimes, we observe a peak at short times (many ephemeral trajectories) and a long-tailed distribution, since trajectories of all durations are possible (Fig. 5d). Even durations longer that our observation time are present, as evidenced by the small peak at 300 s (the duration of this experiment). We next defined a threshold time $t_{th}$

to classify this broad dispersion in trail lengths according to whether their duration from nucleation until merging is shorter or longer than $t_{th}$, calling the short and long-lived trails branches and trees, respectively.

Once the populations are classified according to $t_{th}$, we proceed to analyze the geometry of the spatio-temporal patterns by measuring the equal-time separation of neighboring trails (Fig. 5e). When all trails are included, the distribution features two partially-overlapped peaks. The peak at short scales, $\delta_0$, is indicative of the most likely trail-trail generic distance, while the second maximum, $\delta_{bt}$, identifies a characteristic separation between branches and neighboring trees. By analyzing the distribution of trees only, the distribution is organized around an average value, $\delta_{tt}$, which measures the mutual separation between long-lived trees. The position of the above characteristic lengths depends on the choice of $t_{th}$, which we would tentatively place in the range 25–100 s by looking at the distribution in Fig. 5d. Remarkably, a suitable choice for $t_{th}$ reveals an interesting signature of the kymograph structure. Choosing $t_{th}$ = 50 s, we find $\delta_0$ = 51 ± 10 μm, $\delta_{bt}$ = 112 ± 10 μm, and $\delta_{tt}$ = 205 ± 10 μm, resulting in the approximate relation $\delta_{tt} \approx (\delta_{bt} + 2\delta_0)$. To our understanding, this result illustrates the particular unidimensional regulation of wall defect density that comes to play: while the distance between long-lived defects appears

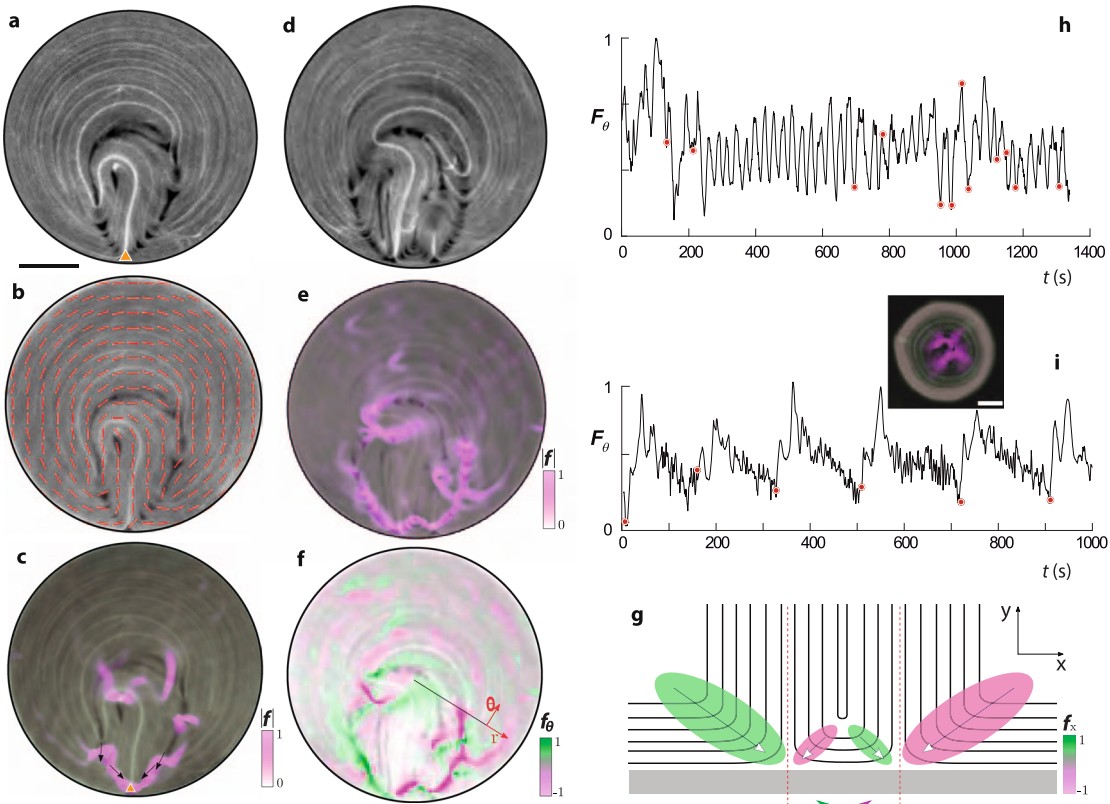

**Fig. 4 | Active forces organized by boundary layer defects. a** Fluorescence micrograph of an active nematic confined in a disk. The orange triangle indicates the location of a wall defect. Scale bar, 100 μm. **b** Corresponding director field overlaid on top of the fluorescence image. **c** Map with the magnitude of the active force per unit area, $f = |\nabla \cdot Q|$, overlaid (in magenta) on top of the fluorescence image. Arrows indicate the direction of the force, while the color intensity maps the magnitude of the force. **d** Fluorescence micrograph of the system with two neighboring wall defects. **e** Magnitude of the active force per unit area overlaid (in magenta) on top of the fluorescence micrograph . **f** Projection of $f$ in the azimuthal

($\theta$) direction. The color code is green for counter-clockwise (CCW) and magenta for clockwise (CW) force. **g** Sketch of the orientational field and the force map in the vicinity of the two neighboring wall defects prior to annihilation. **h** Spatially-integrated azimuthal component of the active force, $F_\theta = \int(e_\theta \cdot \nabla \cdot Q)dr$ as a function of time for the single defect situation (**a–c**). Nucleation events are marked with red disks. **i** Spatially integrated active force as a function of time corresponding to a video from Opathalage et al.[28]. A snapshot from that video and the magnitude of the computed active force is shown as an inset. Scale bar, 50 μm.

regulated by a well-defined wavelength, such a periodic spacing is disrupted by frequent nucleations on both sides of the trees, as described in Section "Structure and dynamics of wall defects". Because these nucleations occur at distances smaller than the natural tree spacing, the resulting defects are necessarily unstable and short-lived.

## Discussion

Our work demonstrates a previously unrecognized role of walls in confined active nematics. Such walls configure active boundary layers that accumulate equal-sign topological defects, pointing to new experimental possibilities of controlling active flows through simple boundary interventions. Interestingly, this scenario is similar to the formation of an electrical double layer surrounding the surface of a polarized electrode in a conductive medium. In that case, adsorbed charges are balanced by a diffuse layer with an excess of charges of the opposite sign. Here, confining walls are polarized by an excess of -1/2 topological charges, while their positive counterparts remain in the bulk.

Our results reveal the existence of a new type of topological defects, which we have called wall defects, whose structure and dynamical behavior are fundamentally different from those of well-known negative bulk defects. We have demonstrated that the existence and nature of those exotic defects is independent of geometrical considerations, since both curved and planar walls behave similarly in this particular respect. At the same time, we have proved the existence

of a true "boundary-layer" effect, since the dynamics close to the wall prevails under strong confinement, and bulk flows can be determined by a pinned wall defect. Our findings reaffirm the potential of engineered boundaries in taming active flows.

Some of the observed behaviors are particularly intriguing case studies in relation to well-established paradigms in ANs. For instance, it is surprising that equal-charge negative defects merge or self-propel, as we have reported here for wall defects. We have shown that this unusual behavior stems from the particular symmetry of wall defects that ultimately renders the boundary layer topologically polarized.

Finally, we provide a statistical description of the structure of the ABL based on the spatial distribution of short-lived and long-lived wall defects. Both the statistical description and the spatiotemporal defect dynamics share qualitative features with pattern-forming systems. In spite of the remarkable similarities between the patterns of localized structures in both systems, the extent of this analogy needs to be assessed with further studies.

## Methods

### Active gel preparation

Microtubules (MTs) were polymerized from heterodimeric $(\alpha, \beta)$-tubulin from bovine brain (Biomaterials Facility, Brandeis University MRSEC, Waltham, MA). The protein was incubated at 37 °C for 30 min in aqueous M2B buffer (80 mM Pipes, 1 mM EGTA, 2 mM MgCl2) prepared with Milli-Q water. The mixture was supplemented with the

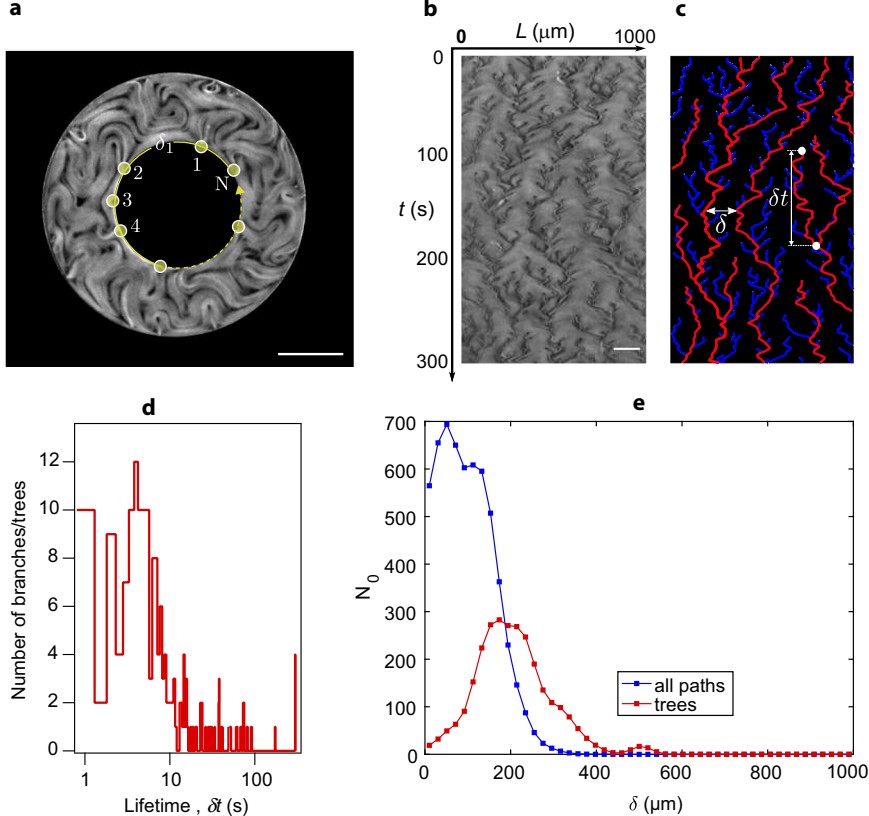

**Fig. 5 | Statistical description of wall defects. a** Wall defects can be regarded as soliton-like features that organize a dynamic distribution of spacings ($\delta_i$) with periodic boundary conditions. **b** Kymograph of the fluorescence intensity of the AN along the inner wall in a 200 μm-wide annulus with an inner radius $R_i = 150$ μm. **c** Skeletonized rendering of the kymograph where long-lived trees have red trajectories while short-lived branches have blue trajectories. The lifetime of a trajectory is $\delta t$, while $\delta$ is the instantaneous distance between two neighboring trees. **d** Lifetime distribution of trees and branches. Statistics in this experiment are obtained over 169 branches and 12 trees. **e** Distribution of distance between neighboring trees and branches combined, and between trees only. $N_0$ is the number of instances of each distance in the kymograph. Here, we have chosen a threshold $\delta_{th} = 50$ s to segregate trees from branches. Scalebars, 150 μm.

reducing agent dithiothrethiol (DTT) (Sigma; 43815) and with guanosine-5-[($\zeta,\beta$)-methyleno]triphosphate (GMPCPP) (Jena Biosciences; NU-405). GMPCPP enhances spontaneous nucleation of MTs, obtaining high-density suspensions of short MTs ($1-2$ μm). 3% of the tubulin was labeled with Alexa 647. Drosophila melanogaster heavy-chain kinesin-1 K401-BCCP-6His was expressed in Escherichia coli using the plasmid WC2 from the Gelles Laboratory (Brandeis University) and purified with a nickel column. After dialysis against 500 mM imidazole aqueous buffer, kinesin concentration was estimated by means of absorption spectroscopy. The protein was stored in a 60% (wt/vol) aqueous sucrose solution at −80 °C for future use. The MTs were mixed with a suspension of kinesin motor clusters (an average of two motors per cluster) in order to act as motile cross-linkers, and Adenosine Triphosphate (ATP) to drive the activity. The non-adsorbing polymer Poly-ethylene glycol (PEG, 20 kDa) promoted the formation of filament bundles through depletion. An enzymatic ATP regenerator system ensured the duration of activity for several hours. A summary of all relevant components of the active gel is summarized in Table 1.

## Active nematic preparation

A PDMS block with a cylindrical well 5 mm in diameter and in depth is glued on a bioinert glass substrated coated with a Poly-acrylamide brush. 2 μL of the active gel is deposited on the substrate and immediately covered with 100 μL of 100 cS silicone oil. A few minutes after this preparation, a AN layer forms at the aqueous/oil interface. A 3D-printed polymeric grid is subsequently submerged in the well using a micromanipulator until it contacts the interface and placed until capillary distortions are minimized (Fig. 6a).

## Grid manufacturing

High-resolution polymeric grids of thickness 100 μm were printed using a Nanoscribe GT Photonic Professional two-photon polymerization printer (Nanoscribe GmbH, Germany) and a 25× objective. The grids were directly printed on silicon substrates without any

**Table 1 | Composition of all stock solutions, and their volume fraction in the final mixture**

| Compound | Stock solution | $v/V_{total}$ |
|---|---|---|
| PEG (20 kDa) | 12 % w vol$^{-1}$ | 0.139 |
| PEP | 200 mM | 0.139 |
| High-salt M2B | 69 mM MgCl$_2$ | 0.05 |
| Trolox | 20 mM | 0.104 |
| ATP | 50 mM | 0.03 |
| Catalase | 3.5 mg ml$^{-1}$ | 0.012 |
| Glucose | 300 mg ml$^{-1}$ | 0.012 |
| Glucose Oxydase | 20 mg ml$^{-1}$ | 0.012 |
| PK/LDH | 600 – 1000 U ml$^{-1}$ | 0.03 |
| DTT | 0.5 M | 0.012 |
| Streptavidin | 0.352 mg ml$^{-1}$ | 0.023 |
| Kinesin | 0.07 mg ml$^{-1}$ | 0.234 |
| Microtubules | 6 mg ml$^{-1}$ | 0.167 |
| Pluronic-147 | 17 % | 0.027 |

*PEG poly-ethylene glycol, PEP phosphoenol pyruvate, M2B buffer solution, ATP adenosin triphosphate, PK/LDH pyruvate kinase/lactic dehydrogenase enzymes, DTT 1,4-dithiothreitol.*

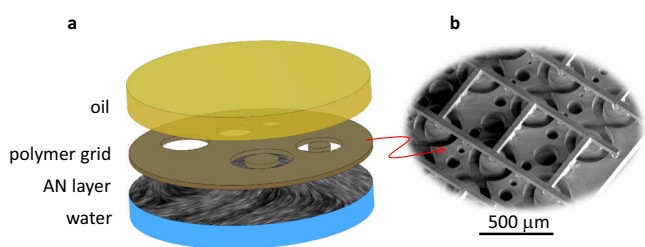

**Fig. 6 | Experimental setup. a** Sketch of channel assembly. The AN layer forms at the water/oil interface inside a cylindrical well. A 3D-printed polymer grid is placed at the interface, thus confining the AN layer. **b** Scanning-electron micrograph of a sample polymer grid.

preparation to avoid adhesion of the resist to the substrate. After printing, the grids are bound to a vertical glass capillary, carefully detached, and washed. Each grid contains different channel geometries so that simultaneous experiments can be performed with the same active nematic preparation, thus ensuring reproducibility (Fig. 6b).

### Imaging and image processing
Images were acquired using a laser scanning confocal microscope Leica TCS SP2 AOBS with a 10 × objective at typical frame rate of 1 image per second. Background correction and kymograph computation were performed with the software ImageJ[43].

The local nematic director, $\boldsymbol{n}$, and nematic tensor, $\boldsymbol{Q}$, where computed using custom Matlab scripts as detailed by Ellis et al.[37]. In brief, the local alignment of the microtubules is inferred using coherence-enhanced diffusion filtering. Noise is first removed from the raw images by means of a Gaussian blur filter with standard deviation $\sigma$. The local molecular director, $\boldsymbol{u}$, is obtained, at the pixel level, by finding the direction along which the fluorescence intensity is most homogeneous. The resulting field $u(x, y)$ is subsequently smoothed by means of a Gaussian blur with standard deviation $\rho$, and used to obtain the local matrices $\boldsymbol{M} = <\boldsymbol{u} \otimes \boldsymbol{u} - \frac{1}{2}\mathbb{1}>_\beta$ using an ensemble average within a disk of radius $\beta$ pixels. These matrices are finally diagonalized choosing, at each pixel, the eigenvector $\boldsymbol{n}$, with the largest eigenvalue, $S$, which will be the nematic director field and order parameter, respectively, and from which the tensor order parameter is computed as $\boldsymbol{Q} = S <\boldsymbol{n} \otimes \boldsymbol{n} - \frac{1}{2}\mathbb{1}>$. The three filtering parameters are manually adjusted by visual inspection of the resulting director field. For the current experiments, we find that typical optimal values are $\sigma = 0.5$px, $\rho = 15$px, and $\beta = 5.5$px.

Once $\boldsymbol{Q}$ is obtained, the local force per unit area is computed as $\boldsymbol{f} \sim \nabla \cdot \boldsymbol{Q}$. Also, the charge density $q$ and the total topological charge $Q$ contained inside a surface $\mathcal{S}$ are computed as $q = \frac{1}{\pi}(\partial_x Q_{x\alpha}\partial_y Q_{y\alpha} - \partial_x Q_{y\alpha}\partial_y Q_{x\alpha})$ and $Q = \int_\mathcal{S} q dS$, respectively[35].

### Data availability
The data that support the findings in this study are available from the corresponding author upon request.

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

## Acknowledgements

The authors are indebted to the Brandeis University MRSEC Biosynthesis facility for providing the tubulin. We thank M. Pons, A. LeRoux, and G. Iruela (Universitat de Barcelona) for their assistance in the expression of motor proteins. We also thank P. Ellis and A. Fernandez-Nieves for kindly sharing all their image processing and defect detection algorithms. J.I.-M., and F.S. acknowledge funding from MICINN (project PID2019-108842GB-C22). J.H. acknowledges funding from the European Union's Horizon 2020 research and innovation program under grant agreement no. 674979-NANOTRANS. T.L.-L. acknowledges funding from the French Agence Nationale de la Recherche (Ref. ANR-13-JS08-006-01) and from the 2015 Grant SESAME MILAMIFAB (Ref. 15013105) for the acquisiton of a Nanoscribe GT Photonic Professional device. Brandeis University MRSEC Biosynthesis facility is supported by NSF MRSEC 2011846. We acknowledge helpful discussions with S. Fraden and M. Norton, on the application of the KS equation with H. Chaté and Olivier Dauchot, and on the topology of wall defects with R. Kamien and O. Lavrentovich.

## Author contributions

J.H. and C.D. performed the experiments, with assistance from J.L. J.H., J.I.-M., and C.D. analyzed the data. F.S and J.I.-M wrote the manuscript with input from all the authors. F.S., J. I.-M., and T.L.-L. conceived and directed the project.

## Competing interests

The authors declare no competing interests.
