## [Peer Review File · Nature Communications]

Active boundary layers in confined active nematicsREVIEWER COMMENTS

Reviewer #1 (Remarks to the Author):

Active boundary layers in confined active nematics:
Jerome Hardouin, et al

Review by Zhenlu Cui

This paper experimentally investigates active boundary layers (ABLs) in a confined active nematic (AN) and shows that a lateral confined AN can develop ABLs.

The authors first study self-assembly of ABLs under annular shaped channels. They employ channels which induce lateral confinement of the quasi-two-dimensional AN, and impose planar anchoring conditions on the ANs and favorable slip velocity conditions along the channel walls. To analyze the process, the authors build kymographs by measuring the fluorescence intensity along circumferences parallel to the boundary. They argue that thickness of ABLs is likely to depend on the active length scale and the channel width, and ABLs are not the result of the known migration of bulk defects towards regions of like curvature sign on curved surfaces.

The authors then consider emergence and interaction between wall defects. They present the process of the birth and death of wall defects. They confine active nematics in corrugated disks with Diameter = 400, 200, 180, 130 μm . They have found that wall defects strongly depend on the disk size and the average number of wall defects decreases when decreasing the disk size, while the indentation acts as a defect attractor where many dynamic branches collapse. They have found a single pinned wall defect when the diameter is decreased below 130 μm . At high concentration of ANs, the system shows periodic oscillations which are in good agreement with previous work by Sanchez et al. [26]. They have evaluated the 2D tensor order parameter and the active forces acting on wall defects.

The authors present and analyze the collective dynamics characterized by the merging tree-shaped patterns and show the existence of wall defects whose structure and dynamical behavior are fundamentally different from those of well-known negative bulk defects. The results are reminiscent of those predicted by the Kuramoto Sivashinski equation (KSE). They have also demonstrated that the existence and nature of wall defects are independent of geometrical considerations.

The authors further provide a statistical description of the structure of ABLs based on the spatial distribution of short-lived and long-lived wall defects. Both the statistical description and the spatiotemporal defect dynamics share qualitative features with pattern-forming systems.

This paper is well-written, clear, and concise and the results are novelty and very interesting. The work demonstrates a previously unrecognized role of walls and ABLs in confined ANs and that the existence and nature of wall defects is independent of geometrical considerations. The work points to new experimental possibilities of controlling active flows through simple boundary interventions. I recommend its publication in Nature Communications. However, I have some minor comments below.

The authors are studying ABLs in a very specific geometrical setting. It might be useful to highlight the physical applications.

ANs are intrinsically different from passive ones in that the activity is introduced into the system, which results many peculiar phenomena such as spontaneous generation, annihilation of defects, instabilities and collective behaviors, etc. It will be nice to provide some comments on it here.

In II A, the authors argue that the thickness of ABLs is likely to depend on the active length scale and the channel width. Is it possible to get more quantitative or qualitative insight into the thickness of ABLs? Are there any mathematical expressions to characterize it? A bit more detail might help here. Indeed, the paper Phys. Rev. E 83 031911 has showed theoretically that the thickness of ABLs is inversely proportional to the square root of the activity.

In IV D, the authors briefly mention how they compute the 2-D tensor order, $Q = S \langle n \otimes n - 1/2 I \rangle$, and the local active stress which strongly depends on the activity. I don't fully understand how the order parameter S and the active stress are computed, although the authors state that the local active stress is computed as $f \sim \nabla \cdot Q$. Are there any mathematical formulas for them?

Reviewer #2 (Remarks to the Author):

The authors present nice experiments, but I am not at all convinced of the significance of the work. Boundary layers, whether active or not, are known to play a major role in many contexts. In conventional liquid crystals, anchoring “enslaves” (as the authors refer to it) and affect the organization in the bulk. In simple fluids, no-slip, for example, causes shear and affects the fluid behavior as a whole. Hence, the fact that what happens at boundaries affects the bulk is not new in any way. Similarly, the lack of a reverse effect is of no real significance, despite what the authors seem to convey. If one has strong tangential anchoring, there is no mechanism for the bulk to change this aspect of the boundary. Hence, the argument that the boundary affects the bulk but that the bulk does not affect the boundary has very little significance. The analogy to spatio-temporal chaos is not at all developed. What do we learn from this analogy about the active nematic? What variables in the model map onto what variables in the active nematic? The paper does not really explain the rationale behind this in any detail that allows appreciating the analogy and/or learning anything meaningful. In addition, the paper is not really self-contained. There are references to the supplementary materials that are critical for following the paper; this indicates the paper might be much more suited for publication in a journal without length constraints. Why are there more or less of the so called negative wall defects at the boundary in some experiments? What determines this? Why is there an intense “plume of high density fluorescent filaments” emanating from the so called negative wall defects? What is the evidence for the cracks referred to in the paper? How do they know these are cracks and if they are what rigidity to they break? I also think the authors have to more clearly explain what happens at the boundaries. The way they refer to wall defects is not complete. Based on their description in panel 2e, they are non-singular defects. They should explicitly say this. However, their experimental data does not really allow elucidating what happens in the vicinity of these “defects” (there is no fluorescent signal in those regions). They ascribe a “charge” to them that I think is misleading. The edge charge in no way needs to be confined

to a narrow region in the boundary. The π rotation at the boundary could be extended to all of the boundary. And it could in addition be any integer multiple of π . In this sense, they are something very different from the usual defects in nematics; the fact they can attract is in this sense not at all surprising as they are simply the reflection of an edge “charge” that could extend along the boundary. Calling them “new type of topological defects” (conclusions) is farfetched. However, explaining why such an edge charge is confined to small regions of the boundary is interesting, and should be explained. Have the authors checked that changes in the edge “charge” couple to the “charge” in the bulk, which is associated to actual defects? This would strengthen their interpretation, particularly because their data do not seem to allow quantifying what happens in regions near this “boundary defects”. For instance, it would allow ruling out that they have a vector field in the boundary and that the “defects” are walls in this field. I also find that the paper uses many buzzwords, but that these have little meaning or that it is not explained or that it even is misleading. For example, what do the authors mean with a topologically polarized wall (abstract, conclusions)? What symmetry does the wall change (abstract)? Perhaps here they are thinking of vector order rather than nematic, but in this case, their edge” charge” would not at all be related to what they seem to now refer to. What collective dynamics do they refer to (abstract)? How many “wall defects” do you need for you to be able to refer to collective dynamics? They refer to a boundary layer being “largely protected from the bulk” (introduction and other places). Protected in what sense? From what? If you are imposing strong planar anchoring that is what you have. In this aspect, I feel the paper should much more extensively discuss what they do and how they achieve it. This seems essential, as there is prior work on exactly the same system with the same overall confinement but that reports different behavior from that found by the authors. As it is now, there is a reference to another manuscript by the authors, but given the significance and the difference in behavior with respect to prior work, they should extend the discussion on the anchoring aspect. Why are the boundary conditions in ref. 21 “ill-defined” and what did they change here to make it so different? As briefly alluded to before, the section on “collective motion” is hard to follow and opaque. It is unclear what one learns from the mentioned analogy to spatio-temporal chaos. It seems a much thorough presentation would allow appreciating what the authors have done much more clearly. I recommend extending the paper to include much of the is presented in the supplementary materials and discuss more extensively what they do and what they find.

Reviewer #3 (Remarks to the Author):

The authors have experimentally studied and analysed the behaviour of extensile active nematic systems in confined spaces, and particularly the role of trapped defects on the boundary of such systems. Their findings are very interesting and novel and it is my opinion that they will be of high interest to the soft/active matter community.

Of particular interest is their findings that show behaviour of trapped defects that are the opposite of what one would expect these defects to do in the bulk of the, e.g. attractive interaction of defects with negative charge when these are trapped in the boundary. They also properly characterize this as a new phenomenon, showing how forces are distributed around these defects, and showing that their behaviour is not dictated by curvature as in some previous works. They also very reasonably conclude on the possibility of using this confinement to control the behaviour of AN systems through manipulation of boundaries. I would recommend that this article be published.

My only comment/question is on the comment the authors make about the case with a single

wall defect on a small circular box without any pinning indentation. They only mention in passing that this defect either fluctuates around the position where it appears, or it tends to drift either clockwise or anticlockwise. It seems to me this can be a rather interesting case in which chirality spontaneously emerges in the system. Looking at the video of that one case, it seems this can come as a spontaneous breaking of the mirror symmetry of the negative charge wall defect across the whole domain, but it's hard to conclude anything from the video alone. Is the dynamics of such a drifting state clear in any way? Once the system breaks this symmetry, is it maintained? Or does the system fluctuate between CW and ACW chirality with some specific lifetime given by other parameters?

We thank the reviewers for their thorough revision and useful comments that have helped improve the clarity of our manuscript. Below, we include itemized responses to all the queries and criticisms and an account of resulting changes in the revised manuscript. Our responses are included in blue text, for better clarity. Also, changes in the updated manuscript have been highlighted with blue text.

REVIEWER COMMENTS

Reviewer #1 (Remarks to the Author):

Active boundary layers in confined active nematics:
Jerome Hardouin, et al

Review by Zhenlu Cui

This paper experimentally investigates active boundary layers (ABLs) in a confined active nematic (AN) and shows that a lateral confined AN can develop ABLs.

The authors first study self-assembly of ABLs under annular shaped channels. They employ channels which induce lateral confinement of the quasi-two-dimensional AN, and impose planar anchoring conditions on the ANs and favorable slip velocity conditions along the channel walls. To analyze the process, the authors build kymographs by measuring the fluorescence intensity along circumferences parallel to the boundary. They argue that thickness of ABLs is likely to depend on the active length scale and the channel width, and ABLs are not the result of the known migration of bulk defects towards regions of like curvature sign on curved surfaces.

The authors then consider emergence and interaction between wall defects. They present the process of the birth and death of wall defects. They confine active nematics in corrugated disks with Diameter = 400, 200, 180, 130 μm . They have found that wall defects strongly depend on the disk size and the average number of wall defects decreases when decreasing the disk size, while the indentation acts as a defect attractor where many dynamic branches collapse. They have found a single pinned wall defect when the diameter is decreased below 130 μm . At high concentration of ANs, the system shows periodic oscillations which are in good agreement with previous work by Sanchez et al. [26]. They have evaluated the 2D tensor order parameter and the active forces acting on wall defects.

The authors present and analyze the collective dynamics characterized by the merging tree-shaped patterns and show the existence of wall defects whose structure and dynamical behavior are fundamentally different from those of well-known negative bulk defects. The results are reminiscent of those predicted by the Kuramoto Sivashinski equation (KSE). They have also demonstrated that the existence and nature of wall defects are independent of geometrical considerations.

The authors further provide a statistical description of the structure of ABLs based on the spatial distribution of short-lived and long-lived wall defects. Both the statistical description and the spatiotemporal defect dynamics share qualitative features with pattern-forming systems.

This paper is well-written, clear, and concise and the results are novelty and very interesting. The work demonstrates a previously unrecognized role of walls and ABLs in confined ANs and that the existence and nature of wall defects is independent of geometrical considerations. The work points to new experimental possibilities of controlling active flows through simple boundary interventions. I recommend its publication in Nature Communications. However, I have some minor comments below.

We appreciate the reviewer's summary with the main ideas and results from our manuscript, and their recommendation to be published in Nature Communications.

The authors are studying ABLs in a very specific geometrical setting. It might be useful to highlight the physical applications.

The role of ABLs is better appraised when the active nematic is confined in space, for instance, in channels. Open channels would be the design of choice for applications. However, for this fundamental study that focuses on the structure and dynamics of ABL, we have considered closed channels (ring-shaped and disk-shaped) in order to avoid the end-effects present in open channels. Moreover, using ring-shaped channels has allowed us to show that the sign of the channel's curvature has no relevance in the formation of the ABL. Yet, as discussed in the text (and exemplified in Fig. S2 of the revised manuscript), ABLs can be also observed on the flat walls of open regions.

ANs are intrinsically different from passive ones in that the activity is introduced into the system, which results many

peculiar phenomena such as spontaneous generation, annihilation of defects, instabilities and collective behaviors, etc. It will be nice to provide some comments on it here.

We thank the reviewer for this suggestion. Following their advice, we have included an additional paragraph in the introduction to better contextualize the role of defects, contrasting the difference between the active and passive case.

In II A, the authors argue that the thickness of ABLs is likely to depend on the active length scale and the channel width. Is it possible to get more quantitative or qualitative insight into the thickness of ABLs? Are there any mathematical expressions to characterize it? A bit more detail might help here. Indeed, the paper Phys. Rev. E 83 031911 has showed theoretically that the thickness of ABLs is inversely proportional to the square root of the activity.

We thank the reviewer for bringing this work to our attention, which we have read carefully. That paper refers to an active fluid under external forcing in conditions of flow alignment and shows that, for planar anchoring, the effect of boundaries penetrates deeper into the bulk for lower activities (Fig. 4 in the paper). We have failed to find a reference showing that the ABL thickness is inversely proportional to the square root of the activity, but this would be indeed the expected dependence for the active length scale on the activity. Since, in our system, the thickness of the ABL is expected to be set by the active length scale, we expect to have this square root dependence also for the ABL thickness.

Actions taken: We have included a mention of this result and a citation of the suggested article (Ref. 32 in the revised manuscript) in the discussion of Fig. 1.

In IV D, the authors briefly mention how they compute the 2-D tensor order, $Q = S \langle n \otimes n - 1/2 I \rangle$, and the local active stress which strongly depends on the activity. I don't fully understand how the order parameter S and the active stress are computed, although the authors state that the local active stress is computed as $f \sim \nabla \cdot Q$. Are there any mathematical formulas for them?

We thank the reviewer for bringing this to our attention. In section IV we mistakenly called $\nabla \cdot Q$ "active stress". As a matter of fact, this term includes an activity prefactor in models, and it should be properly called "active force", or, more precisely "active force per unit area". We direct the reviewer to ref [3], where one can find a review with the main equations used to describe the dynamics of active nematics.

Therefore, we are computing the local active force per unit area or, rather, a function that is proportional to it, as we do not know the precise value of the activity coefficient (see ref. [3]). Concerning the protocol to determine the 2D tensor order parameter and, in particular, the order parameter S , we briefly outline in Sec. IV. D the algorithm we employed, which is an adaptation of the one developed by Ellis et. al. (Ref. [25]).

Reviewer #2 (Remarks to the Author):

The authors present nice experiments, but I am not at all convinced of the significance of the work.

We thank the reviewer for their useful critical comments, which have helped to improve on the clarity of the manuscript and the message conveyed by our work.

Concerning the significance of the work, we introduce in this manuscript a new paradigm: the existence of an active boundary layer, which is populated with active defects that feature extraordinary symmetries and dynamic rules. These defects are negatively charged and nucleate near the confining wall, regardless of the sign of wall curvature. These defects behave differently from their bulk counterparts. For instance, they feature attraction and coarsening between like-sign defects or a collective dynamic that is amenable to a description in terms of a spatiotemporal chaos with a small number of degrees of freedom. For system sizes comparable to the penetration length of the boundary layer, chaotic bulk flows are regularized by the dynamics of wall defects. We highlight that this is the first mechanism capable of explaining the ordering effect exerted by confinement on **active** fluids, and it will be relevant in the emerging field of active microfluidics.

Boundary layers, whether active or not, are known to play a major role in many contexts. In conventional liquid crystals, anchoring "enslaves" (as the authors refer to it) and affect the organization in the bulk. In simple fluids, no-slip, for example, causes shear and affects the fluid behavior as a whole. Hence, the fact that what happens at boundaries affects the bulk is not new in any way. Similarly, the lack of a reverse effect is of no real significance, despite what the authors seem to convey. If one has strong tangential anchoring, there is no mechanism for the bulk to change this aspect of the boundary. Hence, the argument that the boundary affects the bulk but that the bulk does not affect the boundary has very little significance.

We agree with the reviewer that boundaries impose constraints in any system, and that they usually impact the bulk properties. This is precisely why understanding the effects of these constraints is of crucial importance. We know what strong planar anchoring implies for a conventional nematic liquid crystal. However, we don't know what it implies for active materials, in particular for an active nematic that is self-propelled instead of driven by external forces. In this paper, we elucidate this point. We show the existence of a stable defect structure near the boundaries whose properties are essentially different from those of the bulk. This structure penetrates the bulk to a distance that is comparable to the active length scale and controls the system dynamics in situations of high confinement, which are situations of high interest in the perspective of applications (controlled flow in channels, etc.).

The analogy to spatio-temporal chaos is not at all developed. What do we learn from this analogy about the active nematic? What variables in the model map onto what variables in the active nematic? The paper does not really explain the rationale behind this in any detail that allows appreciating the analogy and/or learning anything meaningful.

We would like to emphasize that this part of the manuscript is exploratory, and it only suggests a potential theoretical framework to characterize the collective behavior of wall defect dynamics. The analogy with methods from the field of spatiotemporal chaos is intriguing but demands further studies to properly assess physical insights. That said, we agree with the reviewer that this part was too succinct in the original manuscript, which could have benefited from a more detailed explanation of the analogy.

In the revised manuscript, we have better detailed this analogy. Beyond the similarity between the spatiotemporal patterns in our kymographs for the ABLs and those in the simulations of the one-dimensional KSE (Fig. S1 in the revised manuscript), Ref. 38 of the revised manuscript describes the numerical study of the KSE along a circumference (periodic boundary conditions). That work describes the dynamics of soliton-like modulations of the field parameter simulated by the KSE, in close analogy with the distribution of wall defects that we describe in the ABL. We have, therefore, extended that analogy to adapt the statistical analysis of the distribution of solitons in Ref. 38 to defects in the ABL.

Actions taken: we have extended the introductory part of Sec. IIC to better contextualize the analogy with the KSE analysis of spatiotemporal chaos to more clearly convey the parallelism between the classical theory and our experimental observations of ABL in active nematics, so that the statistical analysis we present is put into the right context. Moreover, we have included a new section and a new figure in the supplementary material (Fig. S1 in the revised manuscript) presenting additional comparative analysis between both systems to further encourage future exploration of these ideas. We believe these additional data and analysis is better suited as supplementary material, since it is not essential to follow the main discussion in the manuscript.

In addition, the paper is not really self-contained. There are references to the supplementary materials that are critical for following the paper; this indicates the paper might be much more suited for publication in a journal without length constraints.

We thank the reviewer for this suggestion. We have promoted to the main text some of the material found in the SI of the original manuscript. We believe that now the manuscript is self-contained.

Details of changes:

- Fig. 4 now incorporates diagrams with the time evolution of the active force, initially found in Fig. S4 of the original manuscript. This facilitates the discussion of the force distribution in the ABL.
- Fig. 5 has been redesigned, incorporating the diagram with the lifetime distribution of boundary defects originally found in Fig. S6. Now the new figure allows a self-contained discussion of the statistical description of wall defects.

Why are there more or less of the so called negative wall defects at the boundary in some experiments? What determines this?

The texture and flow characteristics of the used active nematic material is determined by an intrinsic length scale, called the active length scale, which is a balance between the rigidity of the filaments and the active stress. In turn, these parameters are the result of a precise interaction among a number of biochemical ingredients, as detailed in Table I. Indeed, experimental uncertainties caused by the small volumes that must be handled result in a significant uncertainty in the resulting active length scale of a given preparation which, because of protein degradation, must be used within 24h of its reconstitution. As a result, experiments with different active preparations can only be compared in a qualitative way, while a quantitative analysis can only be performed within the short period of time enabled by the stability of the active preparations.

For instance, experiments in Fig. 3 are performed simultaneously, with the same active preparation, and a printed device that incorporates multiple circular apertures with different diameters. This allows quantitative comparison. In this example, and looking at panel (a), we see that the active length scale (average distance between bulk defects) is much smaller than the pool size. However, in experiments in Fig. 2, performed with a different active preparation, the active length scale is comparable to the disk diameter, resulting in a less turbulent flow, which enables to accurately analyze the dynamics of the ABL. For a given preparation, the value of the active length scale (average distance between defects) will also be a proxy for the inverse average number of defects in the ABL, with a minor contribution due to manufacturing imperfections of the printed device.

All these considerations notwithstanding, we argue that the morphology, dynamics, and role of the ABL will be the same in all cases, with fine details being only a matter of careful engineering.

Why is there an intense “plume of high density fluorescent filaments” emanating from the so called negative wall defects?

The active material is not homogeneously distributed (for instance, we can observe discrete filaments dispersed in the aqueous phase) and, as such, regions with higher and lower packing density coexist. Moreover, certain flow profiles, such as the counter-vortices that self-organize inside a disk confinement, can lead to local active filament accumulation, which is visualized as a more intense fluorescence signal. This are visualized as “plumes” of active material with higher local density.

What is the evidence for the cracks referred to in the paper? How do they know these are cracks and if they are what rigidity to they break?

We thank the reviewer for bringing this to our attention. We realize that the use of the word “crack” can be confusing since, most of the time, we refer to parallel filaments that bend along a direction. The increased curvature leads to higher spacing between filaments, which is visualized by the appearance of “dark regions”, i.e. local areas of low density of fluorescent filaments, piled up along the bending axis. Only when the bending is very abrupt, filaments are unable to comply with the high curvature and indeed break (and spontaneously heal later). An example can be seen, for instance, in Fig. S4, if we compare panels a and d, which we reproduce here. In panel (a), a crimp is formed along the direction where filament bending curvature is maximal (red dashed line). However, filaments are not broken. In panel d, the same region contains broken filaments.

Actions taken: we have replaced the word “crack” with the word “**crimp**” to represent the texture illustrated in panel (a) above, since it better conveys the actual structure.

I also think the authors have to more clearly explain what happens at the boundaries. The way they refer to wall defects is not complete.

Based on their description in panel 2e, they are non-singular defects. They should explicitly say this. However, their experimental data does not really allow elucidating what happens in the vicinity of these “defects” (there is no fluorescent signal in those regions).

We thank the reviewer for bringing this point to our attention, which indeed needs to be clarified. All the topological defects that we refer to in the manuscript are singular defects. We believe that the misunderstanding is probably due to an error in panel (e) of Fig.2, in which the organization of the microtubule bundles around the defect are represented by rods. Indeed, there is a rod missing in the bottom side of the defect, near the wall (see sketches below). These rods should completely surround the defect for it to be singular, as experimental evidence suggests being the case. We have mended this and we have also removed the small diagram depicting a $-1/2$ edge defect, which was confusing.

Actions taken: we have corrected panel (e) in Fig. 2.

They ascribe a “charge” to them that I think is misleading. The edge charge in no way needs to be confined to a narrow region in the boundary. The π rotation at the boundary could be extended to all of the boundary. And it could in addition be any integer multiple of π . In this sense, they are something very different from the usual defects in nematics; the fact they can attract is in this sense not at all surprising as they are simply the reflection of an edge “charge” that could extend along the boundary. Calling them “new type of topological defects” (conclusions) is farfetched. However, explaining why such an edge charge is confined to small regions of the boundary is interesting, and should be explained. Have the authors checked that changes in the edge “charge” couple to the “charge” in the bulk, which is associated to actual defects? This would strengthen their interpretation, particularly because their data do not seem to allow quantifying what happens in regions near this “boundary defects”. For instance, it would allow ruling out that they have a vector field in the boundary and that the “defects” are walls in this field.

We thank the reviewer for bringing this issue to our attention. We used the term “edge charge” as a synonym of “the charge of a wall defect”. We realize that this word may lead to confusion, so we have removed “edge charge” everywhere it appeared and replaced it with “the charge of wall defects”.

The topological charge we are referring to is the winding number of the director field at the defect, as in passive liquid crystals. It is a local property. This definition also holds for defects nucleating at walls, independently of the wall geometry. For instance, passive nematic liquid crystals confined by sawtooth walls with different angles lead to the formation of disclinations with non-half-integer charges, always localized at the apex of the distortion [see, for instance, Phys. Rev. E **82**, 011707 (2010)].

In our system, we can obtain the topological charge of defects, either directly, by examining the director field around them, or indirectly, by taking into account that the total topological charge of the system needs to be conserved. Indeed, the value $s=-1/2$ assigned to the topological charge of wall defects can be justified by analyzing their interaction with bulk defects. For instance, in Fig. 2d we sketch an instance of defect nucleation and unbinding (experimental snapshots in Fig. 2c). The parallel extensile active filaments become unstable, and they buckle, leading to the unbinding of a defect pair. One of the defects is injected into the bulk and its structure is clearly that of a $+1/2$ bulk defect. From here, charge conservation indicates that the defect that remains always in the ABL must have a topological charge $-1/2$.

Another example is afforded by Fig. 2h where we sketch the “annihilation of two boundary defects” (experimental snapshots in Fig. 2g). Two boundary defects approach, squeezing a bulk $+1/2$ defects between them. The result is a single boundary defect. Once again, charge balance indicates that the topological charge of boundary defects must be $-1/2$.

Actions taken: we have replaced the term “edge charge” with “charge” in reference to defects in the ABL.

I also find that the paper uses many buzzwords, but that these have little meaning or that it is not explained or that it even is misleading. For example, what do the authors mean with a topologically polarized wall (abstract, conclusions)?

We share this reviewer’s view that pointless use of buzzwords should be avoided. Nevertheless, we believe that the expression “topologically polarized wall” has an important meaning that, perhaps, we failed to properly convey. In this work, we present this new active structure, that we call active boundary layer, that is populated by identical topological defects of charge $+1/2$. Our view was to make an analogy with, for instance, the electrical double layer

surrounding colloidal particles in suspension. The particle may have a negative surface charge (for instance, because it contains ionizable surface moieties with a fixed anion) and will be surrounded by a diffuse region with an excess positive electrostatic charge density to have a net charge balance. The particle is said to be polarized, as its surface electrostatic potential is different from that in the bulk.

In our case, ABL's are populated by $s=-1/2$ topological charges. We can thus extend the above language to say that ABL are polarized (since they harbor a net negative charge, when compared with the bulk that should include an excess positive charge). Since we are talking about topological, not electrostatic, charges, then we say that ABL are topologically charged walls.

Actions taken: we have explained this "topological charge polarization" in a new paragraph in the introduction, and we have better explained the analogy with the electrical double layer in the conclusions paragraph.

What symmetry does the wall change (abstract)? Perhaps here they are thinking of vector order rather than nematic, but in this case, their edge" charge" would not at all be related to what they seem to now refer to.

Wall defects are polar (head towards the bulk) and have a bilateral symmetry while bulk $-1/2$ defects are apolar have a three-fold symmetry. Bulk defects, with a three-fold symmetry, are only advected by the flow in the active material. They are not self-propelled, on average. In the case of wall defects (even when there is a single one) defects are self-propelled when their bilateral symmetry breaks. However, due to their polarity, they are constrained to move along the wall. As sketched in Fig. 2, panels e and f, this symmetry is also translated in the different symmetry of active stresses. In Fig. 4, panels a-f, we have actually estimated the active stresses in the simple situation of just one or two boundary defects, showing how an imbalance in the lateral symmetry of active stress leads to the propulsion of defects in the ABL.

What collective dynamics do they refer to (abstract)? How many "wall defects" do you need for you to be able to refer to collective dynamics?

When we analyze the wall defects in the ABL, we focus on the birth of individual defects and in their pairwise annihilation. However, ABLs such as those in the ring channel shown in Figure 1 feature a roughly steady number of wall defects that evolve with a seemingly chaotic dynamics. As described in Sec. IIC (extended in the revised manuscript), we propose an analogy with the dynamics of soliton-like structures in spatiotemporal chaos. Certainly, the number of defects in a given ABL is limited, and the proposed statistical description should improve as that number increase.

They refer to a boundary layer being "largely protected from the bulk" (introduction and other places). Protected in what sense? From what? If you are imposing strong planar anchoring that is what you have.

We thank the reviewer for this question. By "protected from the bulk" we mean that the ABL features an average, steady-state density of wall defects that is maintained in spite of its interactions with the "bulk". Interestingly, this is similar to the observations presented in Ref. 29 of the revised manuscript, where the AN is prepared over a surface where steps and terraces have been machined. The authors report a topological charge separation and an accumulation of $-1/2$ defects over step edges with a steady-state density, although in that case defects are regular bulk $-1/2$ defects. In our experiments, this "protection" can be taken further by incorporating pinning of $-1/2$ with a small indentation, which such defects to be static and never annihilated.

We would like to emphasize that the concept of strong/weak planar anchoring, which is well-known in passive liquid crystals has still unknown implications in the case of active systems. Please, see our comment above regarding the significance of this work.

In this aspect, I feel the paper should much more extensively discuss what they do and how they achieve it. This seems essential, as there is prior work on exactly the same system with the same overall confinement but that reports different behavior from that found by the authors. As it is now, there is a reference to another manuscript by the authors, but given the significance and the difference in behavior with respect to prior work, they should extend the discussion on the anchoring aspect. Why are the boundary conditions in ref. 21 "ill-defined" and what did they change here to make it so different?

We thank the reviewer for these comments. We agree that the comparison/contrast with earlier work should be clear. In private communications with the authors of Opatalage et al. (Ref. 21 in the original manuscript, Ref. 28 in the revised manuscript), it has been made clear that their experimental preparation hindered a proper study of the boundary region. In their case, the active nematic formed over a substrate that contained oil-covered wells. This

created a boundary that is curved in the third dimension, preventing a correct focus and a consistent analysis. Therefore, they restricted their study to the bulk dynamics of confined active nematics. In some recent data that the authors of Ref. 28 have shared with us, where (by chance) the interface is in focus, one can measure a kymograph that is not unlike the ones we report here. In our experiments, the confining grids are applied on an already-formed active nematic, which results in more consistent boundary structures, that lay on a perfectly flat interface.

In summary, active boundary layers were likely at play in earlier experiments in Ref. 28, but perhaps not in all their preparations. In particular, the dynamics of AN inside a disk that we discuss in Sec. B3 is markedly different in our experiments and in Ref. 28, where it very much resembles the dynamics in the presence of weak confinement that we observed earlier when the AN evolved over substrates with circular domains and that has recently been reported by Linda Hirst in AN evolving over pillars (Ref. 35 in the revised manuscript). All these evidences suggest that lateral confinement was not very consistent in the experiments of Ref. 28.

On the other hand, one source of confusion when comparing our work with that in Ref. 28, acknowledged in a private communication by the authors of that work, is the nature of the boundary conditions. Early in Ref. 28 they claim that hard walls impose no-slip on the AN, as this is also the prescription for their models. Perhaps the water has stick boundary conditions, but it is clear, when looking at their unpublished experiments where the boundary is better resolved, that active filaments have a significant slip along the boundaries. This is what we meant when writing that boundary conditions in Ref. 28 were “ill-defined”, but we agree that such statement needs clarification and substantiation, and we have chosen to rewrite our comparative discussion since it should be based on data published in Ref. 28 and not on private communications.

Actions taken: we have rewritten the final paragraph in Sec. B3 as follows:

“Interestingly, a different dynamics is observed in the experiments by Opathalage et al. [28] on an AN inside circular cavities. In their experiments, a pair of central evolving $+1/2$ defects are periodically disturbed by the nucleation of a couple of oppositely charged defects. The new positive defect further unbinds and replaces one of the original central $+1/2$ defects, which annihilates with the newly created negative defect (Fig. 4i). Minima of the stress occur just before defect nucleation. This periodic behavior is reminiscent of the observations in unconfined or weakly confined ANs [15,16,35], and different from our system whose dynamics is controlled by ABLs. While, in our work, lateral confinement is applied on an already-formed AN, experiments in Ref. [28] are performed by assembling the AN on a patterned, oil-coated substrate. The latter results in the formation of menisci along the pool boundaries, which may lead to interfaces that deform along the normal direction, thus preventing observation of the AN with a good resolution.”

As briefly alluded to before, the section on “collective motion” is hard to follow and opaque. It is unclear what one learns from the mentioned analogy to spatio-temporal chaos. It seems a much thorough presentation would allow appreciating what the authors have done much more clearly. I recommend extending the paper to include much of the is presented in the supplementary materials and discuss more extensively what they do and what they find.

We thank the reviewer for this suggestion. As commented above, this part of the study is an intriguing, yet tentative analogy with the collective dynamics of soliton-like structures in spatiotemporal chaos, and would benefit for further study, which is beyond the main scope of this work. Interestingly, this is not the first time the dynamics of active nematics is related to classical concepts of statistical physics. In a recent publication (Ref. 25 in the revised manuscript), some of us showed that the onset of the active turbulence regime in unconfined active nematics can be described as a genuine selection mechanism, not unlike those found in classical pattern formation.

Actions taken: we have extended the introductory part of Sec. IIC to better contextualize the analogy with the KSE of spatiotemporal chaos to more clearly convey the parallelism between the classical theory and our experimental observations in ABL in active nematics, so that the statistical analysis we present is put in the right context. Moreover, we have included a new section and a new figure in the supplementary material (Fig. S1 in the revised manuscript) showing additional comparative analysis between both systems to further encourage future exploration of these ideas. We believe this additional data and analysis is better suited as supplementary material, since it is not essential to follow the main discussion in the manuscript.

Reviewer #3 (Remarks to the Author):

The authors have experimentally studied and analysed the behaviour of extensile active nematic systems in confined spaces, and particularly the role of trapped defects on the boundary of such systems. Their findings are very interesting and novel and it is my opinion that they will be of high interest to the soft/active matter community.

Of particular interest is their findings that show behaviour of trapped defects that are the opposite of what one would expect these defects to do in the bulk of the, e.g. attractive interaction of defects with negative charge when these are trapped in the boundary. They also properly characterize this as a new phenomenon, showing how forces are distributed around these defects, and showing that their behaviour is not dictated by curvature as in some previous works. They also very reasonably conclude on the possibility of using this confinement to control the behaviour of AN systems through manipulation of boundaries. I would recommend that this article be published.

We thank the reviewer for their positive comments and the recommendation that the manuscript is accepted.

My only comment/question is on the comment the authors make about the case with a single wall defect on a small circular box without any pinning indentation. They only mention in passing that this defect either fluctuates around the position where it appears, or it tends to drift either clockwise or anticlockwise. It seems to me this can be a rather interesting case in which chirality spontaneously emerges in the system. Looking at the video of that one case, it seems this can come as a spontaneous breaking of the mirror symmetry of the negative charge wall defect across the whole domain, but it's hard to conclude anything from the video alone. Is the dynamics of such a drifting state clear in any way? Once the system breaks this symmetry, is it maintained? Or does the system fluctuate between CW and ACW chirality with some specific lifetime given by other parameters?

This is an intriguing idea raised by the reviewer. Due to experimental constraints, our observations are time-limited. In these periods of time, we observe that the drift takes a random handedness, as expected in the absence of external symmetry breaking. Such motion fluctuates with time, but we have failed to observe episodes of handedness inversion. Perhaps longer experiments would reveal such events. Nevertheless, in separate experiments we have seen that roughening the boundary with ratchet-like structures can determine this drifting handedness. One should expect the boundary to always contain certain micro-roughness, so this may lead to uncontrolled symmetry breaking in the experiments presented in this manuscript.

REVIEWER COMMENTS

Reviewer #1 (Remarks to the Author):

The authors have done an excellent revision from their earlier version. From a theoretical point of view, all my concerns including the physical applications for the specific geometrical setting, the role of defects and the difference between the active and passive case, the effect of the activity, and the computation of the local active force, were well addressed.

Here I would like to give a brief explanation on the theoretical prediction on the thickness of the boundary layer in Ref. #32: Since the dominating terms in Equ (26) near $y = \pm 1$ are $e^{\delta y}/e^{\delta r}$, when the activity δ (For extensile materials: $\delta < 0$, and contractile materials: $\delta > 0$) is strong, the steady states have a boundary layer near the wall, whose width is proportional to $1/\sqrt{|\delta|}$.

I recommend it for publication.

Reviewer #2 (Remarks to the Author):

I have read with interest the response provided by the authors to the referee comments, as well as the new version of the manuscript. I appreciate the effort of the authors. I find the manuscript has improved and that it is now clearer. Still, I think there are aspects that need further clarification. The authors state that "certain flow profiles can lead to local active filament accumulation". Do flow profiles in experiment explain the observed plumes? I find the schematic 2(e) to be quite confusing, as it would imply the defect is a bulk $-1/2$ defect close to the wall. Is this what the authors think they have? How do they know if there are almost no filaments in the images near these defects? Assuming they can ascertain experimentally this is the case, which they should, the schematics in 2(d) and 2(h) would be misleading, as these show the defect as truly lying on the boundary. Can the authors demonstrate that the total topological charge is consistent with the topological properties of the surface where their defects live, which has boundaries? This can shed light on how to really think about these defects. The authors response to some of my comments in this respect were not completely addressed, and I believe doing this can only give them arguments to justify what they seem to claim or to revise their claims appropriately. This is important, since these "wall" defects are in large part the focus of the work. I also feel that the comment on curvature is highly misleading; the attraction of positive and negative defects to regions with positive and negative Gaussian curvature pertains to surface curvature. In this respect, their surface is flat and not at all curved. It has curved boundaries, but these are characterized by the curvature of the associated curve, which is distinct to the surface curvature alluded to. I urge the authors to revise their statements based on this fact. I now better see the analogy to a polarized interface. In this respect, while the example in the response refers to a charged colloid, which is not truly polarized as a whole due to the spherical symmetry of the electric double layer, in the paper they refer to an electrode, where indeed a net polarization is observed. I therefore appreciate (and like) the analogy made by the authors in the manuscript. Finally, I also believe the comparison between their work and prior work is made clearer in the new version of the manuscript. As it is the comparison and rationale to the KS model. Once the authors address my comments above in a convincing way, I would be happy to recommend publication of their manuscript.

Reviewer #3 (Remarks to the Author):

I am satisfied on my side with the authors response and I think this manuscript should be published. The dynamics of defects in active mechanics has attracted a lot of attention in recent years. Active nematic interfaces have attracted quite a lot of attention recently and their stability and dynamics depends quite a lot on how defects behave close to these interfaces.

It seems for instance interfaces between active nematics and other media impose anchoring (rather unsurprisingly) through either geometric alignment or flow interactions. It's been observed however that the appearance and behaviour of defects in these interfaces dictate then the behaviour of said interface, pointing to different processes like instabilities or traveling waves. Thus understanding of this behaviour close to walls offers a simpler route to study this defects dynamics and might be of use and interest to those trying to understand these interface and boundary phenomena.

While the particular use of these results is hard to gauge, since these interfaces hasnt been studied in a lot of detail, yet, (see for example [1]), and thus the appropriateness of this particular journal for the dissemination of this work is not completely clear, I still think these are interesting and important results and many people will find them relevant. It might as well motivate some analytical study of these wall dynamics in more detail.

Blow, Thampi & Yeomans, Phys Rev Lett 113 248303

We thank the reviewers for their thorough revision and useful comments. We appreciate the suggestions by reviewer 1 and 3 that our manuscript be accepted for publication in its current form. Here we provide an itemized answer to reviewer's 2 remaining questions and concerns. Our responses are included here in blue text, for better clarity. Also, changes in the updated manuscript have been highlighted with blue text.

I have read with interest the response provided by the authors to the referee comments, as well as the new version of the manuscript. I appreciate the effort of the authors. I find the manuscript has improved and that it is now clearer. Still, I think there are aspects that need further clarification.

The authors state that "certain flow profiles can lead to local active filament accumulation". Do flow profiles in experiment explain the observed plumes?

We thank the reviewer for this question. We realize we did not properly discuss the flow profiles that, we believe, are responsible for these "plumes".

Indeed, if one looks at Movie S2 (which corresponds to an experiment inside a disk-shaped pool featuring a minimal number of wall defects), a laminar flow profile extends over most part of the surface except for a region between the "main" wall defect and the center of the pool. Overall, the flow is organized as two counter-rotating vortices, whose velocity is parallel to the solid boundaries. This flow pattern is quite apparent when visualizing the videos. Measuring the actual flow profile is quite challenging, in particular close to the walls (due to the high homogeneity of the material). We have tested a recently developed algorithm, called *BioFlow*, that allows us to extract a quite accurate map of this flow profile, as shown in the sketch (arrow lengths are proportional to local speeds):

The flow is nearly symmetric about an axis defined by the main wall defect and the center of the pool. The two vortices converge at the tip of the wall defect, effectively concentrating the fluorescent filaments of the active material into a thicker and brighter filament. This forms the bright plume that is advected by the flow. As we mentioned in section B2, similar structures were reported a few years ago by Dogic and co-workers [Science 2011, Ref. 36 in the revised manuscript]. These authors described "cilia-like" active filaments that formed at the boundaries of a bulk active gel.

Actions taken: To clarify these ideas, we have extended the corresponding discussion in Sec. B.1 as follows:

The defect core is prolonged by a plume of high-density fluorescent filaments (Fig. 2a), reminiscent of the beating active filaments reported by Sanchez et al. at the boundary of a bulk active gel [34]. Inspection of the quasi-laminar flow profiles inside the pool (see Movie S2) reveals two steady counter-rotating vortices symmetrically organized on either side of the wall defect with velocities parallel to the pool boundary. These flows converge at the tip of the wall defect, effectively concentrating the fluorescent active filaments, which results in a bright plume that is advected by the flows.

I find the schematic 2(e) to be quite confusing, as it would imply the defect is a bulk $-1/2$ defect close to the wall. Is this what the authors think they have? How do they know if there are almost no filaments in the images near these defects? Assuming they can ascertain experimentally this is the case, which they should, the schematics in 2(d) and 2(h) would be misleading, as these show the defect as truly lying on the boundary.

We agree with the reviewer that those schematics must be clarified, and better discussed. The nature of wall defects and whether there are filaments between the core of the defect and the wall is relevant to understand why they remain at the wall region and are not injected into the bulk, but it is not so important to quantify their topological charge. After careful inspection of high-resolution confocal imaging (Movie. S4 in the revised manuscript), we are confident that there are no filaments between the core of the defect (devoid of fluorescent filaments) and the wall. Any band of filaments behind the defect core would eventually be unstable and feature a bend instability that would push the wall defect into the bulk. This does not happen to stable wall defects.

ABL defects are bulk defects sitting on a wall and, as such, they have both bulk charge (associated to the distortions that they induce in the two-dimensional orientational field) and edge charge (associated to the distortions they induce along the wall). Upon defect unbinding (Fig. 2c, 2d), topological charge conservation indicates that a wall defect must have a bulk charge of $-1/2$ to compensate for the $+1/2$ injected into the bulk. Likewise, one can also associate an edge charge to each wall defect by analyzing the intersection of the orientational field and the one-dimensional boundary. In this case, the director undergoes a π -rotation when crossing the defect, so the edge charge is also $-1/2$.

This same idea can be discussed regarding three-dimensional defects caused by the inclusion of colloidal spheres in a uniform nematic field. Let us consider the double boojum configuration often obtained for planar anchoring (see diagram). This configuration creates one $+1$ surface defect or boojum at each pole of the particle. However, each of these defects also creates distortions in the bulk, so they have associated a three-dimensional bulk charge. Specifically, they are half hyperbolic hedgehogs. In this case, the bulk charge and surface charge associated to the defect are different. In ABL, in contrast, defect bulk charge and edge charge coincide, as both are $-1/2$.

This subtlety between edge charge and bulk charge is not relevant to understand the dynamics of wall defects and may lead non-specialists to confusion. In this manuscript, we are concerned about how ABL defects interact with the bulk, and thus, we should focus on their bulk properties (bulk charge).

Actions taken: we have revised Fig. 2 in the manuscript to illustrate that the core of wall defects reaches the boundary. We have thoroughly modified the text in Sec. B.I to clearly convey the above ideas. We have also acknowledged helpful discussion regarding these issues with O. Lavrentovich and R. Kamien during the International Liquid Crystal Conference.

Can the authors demonstrate that the total topological charge is consistent with the topological properties of the surface where their defects live, which has boundaries?

We thank the reviewer for this suggestion, as this is an interesting cross-check. Indeed, the total bulk charge inside a disk with planar anchoring conditions on the boundary is expected to be $+1$, regardless of the complex internal defect dynamics. To compute this charge, we have followed the expressions derived by Blow, Thampi, and Yeomans (new Ref. 35 in the revised manuscript) to first

compute the local charge density associated to the tensorial order parameter of the active nematic. We have included the corresponding “density map” for the configuration shown in Fig. 2a as the new panel Fig. 2c, which we reproduce here. We have also included a new movie, Movie S3 in the revised manuscript, where the evolution of the charge density is monitored for 25 seconds.

To compute the total charge, we integrate the charge density over the disk surface. We have included the new panel Fig. 2d where we plot the evolution of the total charge with time, and show that its value is +1, as expected.

Actions taken. We have included two new panels in Fig. 1 with the computation of the charge density and the time evolution of the total charge inside the disk-shaped domain. We have also included an additional video, we have discussed the new panels in the text, and we have detailed how these new computations are performed in the Materials and Methods.

This can shed light on how to really think about these defects. The authors response to some of my comments in this respect were not completely addressed, and I believe doing this can only give them arguments to justify what they seem to claim or to revise their claims appropriately. This is important, since these “wall” defects are in large part te focus of the work.

We thank the reviewer for insisting on this aspect of the manuscript, as it is indeed crucial. We trust that the explanations above and the changes in the manuscript have contributed to convey a clear message. As we have shown, a computation of the total charge inside a disk-shaped domain yields a value +1 at all times, coherent with boundary conditions and with the dynamics of boundary defects that we describe in the manuscript.

I also feel that the comment on curvature is highly misleading; the attraction of positive and negative defects to regions with positive and negative Gaussian curvature pertains to surface curvature. In this respect, their surface is flat and not at all curved. It has curved boundaries, but these are characterized by the curvature of the associated curve, which is distinct to the surface curvature alluded to. I urge the authors to revise their statements based on this fact.

We thank the reviewer for this observation. We agree that the two scenarios are very different. In the original manuscript, we wanted to contrast the observed preference of wall defects for boundaries of all shapes with the migration of negative defects to regions of negative Gaussian curvature on surfaces. We realize that any reference to the latter known phenomenon on surfaces may lead to some degree of confusion here, where the interface is flat and, since it does not help to clarify our ABL structure, we have opted to remove the comparison, along with ref. 34 and 40 in the original manuscript (notice that we keep Ellis et al, ref. 37 in the revised manuscript, as we refer to it in the Methods section).

Actions taken: We have modified the discussion in section II.A and have removed the initial sentence in the second paragraph of Sec. III.

With this, the last sentence at the end of Section II.A now reads:

The situation is similar at the inner and at the outer annulus walls and also near flat walls in rectangular channels (Fig. S2), indicating that ABLs form regardless of the channel geometry.

I now better see the analogy to a polarize interface.

In this respect, while the example in the response refers to a charged colloid, which is not truly polarized as a whole due the spherical symmetry of the electric double layer, in the paper they refer to an electrode, where indeed a net polarization is observed. I therefore appreciate (and like) the analogy made by the authors in the manuscript.

Finally, I also believe the comparison between their work and prior work is made clearer in the new version of the manuscript. As it is the comparison and rationale to the KS model.

Once the authors address my comments above in a convincing way, I would be happy to recommend publication of their manuscript.

We thank this reviewer once more for the thorough revision and useful comments that have helped us to significantly improve the clarity of the original manuscript. We trust that the additional changes and explanations have helped to clarify all pending issues.

REVIEWERS' COMMENTS

Reviewer #2 (Remarks to the Author):

I believe the authors have addressed all of my concerns and that the paper is now more precise and the results more clearly presented. As a result, I recommend publication.